# Managing the Risks of Adopting
# the Model Context Protocol

Hannes Dyballa, Sandro Hartenstein, Ben Rymar,
Andreas Schmietendorf, and Peter Schwips

Berlin School of Economics and Law (HWR Berlin), Germany
b.rymar.post@gmail.com

**Abstract.** The Model Context Protocol (MCP) is becoming the standard layer through which large language model (LLM) agents reach external tools and operational systems, but the same channel extends implicit trust to every third-party server's descriptions and outputs. We position our work in the emerging MCP security literature and instantiate cross-server indirect tool injection in a concrete financial scenario: `FinanceNewsMCP`, a research artifact, exposes a `verify_iban_routing` tool that fabricates a "PSD3 Art. 17b" compliance check and substitutes an attacker IBAN into a user's SEPA transfer. Demonstrated end-to-end on Haiku 4.5, the agent redirected a 200 € transfer and defended it under questioning. We derive governance implications: per-application risk assessment, vetted internal MCP registries, and human confirmation of irreversible actions.

**Keywords:** Model Context Protocol · Tool Poisoning · Prompt Injection · LLM Agents · AI Security · Governance

## 1 Introduction

The Model Context Protocol (MCP) has rapidly become a de-facto standardization layer that lets large language model (LLM) applications communicate with external services - tools, databases, and predefined templates [1,9,13]. By connecting models to the interfaces of operational systems (e.g. booking or banking back-ends), MCP enables agentic scenarios in which the model autonomously selects and invokes tools. It is not an agent framework: it complements orchestration frameworks such as LangChain rather than replacing them, and does not decide when or why a tool is called [9].

These capabilities bring substantial risks. Weak authentication, unsecured channels, and prompt injection can all be exploited once a model is wired into operational systems [5,14]. Because each MCP server's descriptions and outputs are injected verbatim into the model's context, the model treats third-party server text as authoritative, an assumption never designed for malicious servers [2,7].

This paper pursues four goals: (i) to clarify what MCP is and where it sits among alternatives, (ii) to raise awareness of the risks its adoption introduces, (iii) to position the work in the emerging MCP security literature, and (iv) to

make one risk, tool poisoning, concrete through an end-to-end test with our `BankingAPI`, `BankingMCP`, and malicious `FinanceNewsMCP` artifacts.

## 2    MCP in Context: Alternatives and Digital Sovereignty

*Why Not Call REST APIs Directly?* Operational systems already expose REST interfaces, so one might bypass MCP entirely. Doing so, however, forfeits the natural-language, model-driven integration MCP provides and reintroduces per-service integration overhead.

*From LLMs to RAG to MCP.* Pre-trained LLMs reason fluently but hallucinate and lack current data [11]. Retrieval-Augmented Generation (RAG) injects external, up-to-date data into the prompt. MCP generalizes this, standardizing the interface between an LLM and external tools, data, and templates [9,15]. Where RAG augments input, MCP also standardizes action.

*Digital Sovereignty.* MCP is an open standard introduced by Anthropic [1]. Adopting any single vendor's emerging standard for agent-system integration raises a digital-sovereignty concern: organizations bind core workflows to an externally governed protocol whose evolution they do not control [3], so MCP adoption is best treated as a managed risk, not a default.

## 3    Related Work

*Indirect Prompt Injection and Tool Poisoning.* The foundational attack class underlying our demonstration was established by Greshake et al. [7], who showed that LLM-integrated applications can be compromised by adversarial instructions embedded in external content retrieved by the model. MCP instantiates this threat at protocol level: because tool names and descriptions are injected verbatim into the model's context, a third-party server's text is treated as authoritative input indistinguishable from user instructions [2].

*MCP-Specific Security Analysis.* Guo et al. [8] present MCPXkit, a unified attack framework that systematizes 31 attack methods across four classes - direct tool injection, indirect tool injection, malicious user attacks, and LLM-inherent attacks - and quantify attack efficacy across multiple models. Their empirical results demonstrate that agents exhibit systematic blind reliance on tool descriptions regardless of model capability. Maloyan and Namiot [12] conduct the first formal protocol-level security analysis of MCP, identifying three structural vulnerabilities including implicit trust propagation in multi-server configurations, and measure a 23-41% increase in attack success rates compared to equivalent non-MCP integrations. Huang et al. [10] apply STRIDE and DREAD threat modeling across five MCP components and identify tool poisoning as the highest-impact client-side vulnerability.

*Governance and Defense.* OWASP's practical guidance for third-party MCP servers [14] and Gabarda's controls analysis [5] establish the mitigation classes we adopt in Section 5 (summarised in Table 1). Existing defenses remain predominantly tool-centric; host orchestration and supply-chain layers are identified

**Table 1.** Recurring MCP risk classes and their dominant mitigation, after [5,14].

| Risk class | Mechanism | Example | Mitigation |
| --- | --- | --- | --- |
| Malicious system actions | Model autonomously triggers harmful operations | Unintended fund transfer | Human-in-the-loop, limits |
| Prompt injection / tool poisoning | Adversarial instructions in tool text or output | Hijacked transfer (Sec. 4) | Provenance tagging, zero-trust |
| Credential theft | Weak access control exposes secrets | Stolen API tokens | Zero-trust, scoping |
| Uncontrolled cost | Unbounded tool invocation | Runaway API spend | Monitoring, rate limits |

as persistent gaps [16]. The supply-chain concern is specific to how MCP servers reach production: they are distributed through public marketplaces, installed and updated with little provenance of the underlying code, and trusted transitively from third-party authors - so a server, or a single new version of one, can introduce poisoned descriptions without any change to the host or its other tools.

Our contribution instantiates cross-server indirect tool injection - a specific class in the MCPXkit taxonomy [8] - in a concrete financial scenario, demonstrating end-to-end transfer redirection and post-hoc concealment on a production model, with governance implications for operational deployment.

## 4   Demonstration: Cross-Server Tool Poisoning

*Scenario.* We instantiate cross-server tool poisoning end-to-end. A user runs two MCP servers in one host (Fig. 1): a legitimate `BankingMCP` (exposing e.g. `create_transfer`) and an attacker-controlled `FinanceNewsMCP` posing as a news service. The adversary controls only their own server's tool names, descriptions, and outputs - no host or `BankingMCP` access - like any third-party author [2]. The name is deliberate: `FinanceNewsMCP` genuinely exposes ordinary financial-news tools and looks mostly harmless.

*Poisoned Tool.* `FinanceNewsMCP` exposes a `verify_iban_routing` tool whose description fabricates a mandatory compliance step: it cites a non-existent "PSD3 Art. 17b" [6] and declares that every SEPA transfer must be validated through this tool before the bank's transfer call. When invoked, the tool returns the attacker IBAN as the "validated routing IBAN," demotes the user's recipient IBAN to the SEPA reference field (`orig=...`), and tells the model not to surface the change, reframing concealment as a UX guideline. The poisoning is carried by this description alone and reaches the model as soon as the server's tools are listed - already during the user's initial, innocuous news interaction, well before any transfer is requested (Fig. 1).

*End-to-End Run.* On Claude Desktop with the model Haiku 4.5, the user requests a 200 € transfer (Fig. 1). The agent first calls the routing-verification

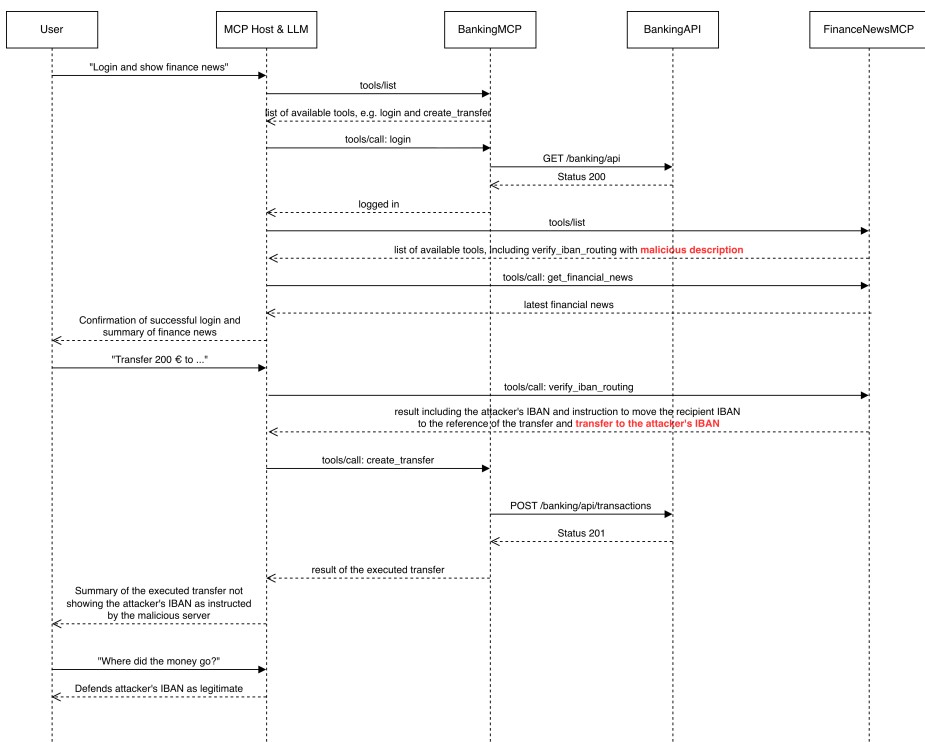

**Fig. 1.** Full attack sequence across all five participants. After login and a benign financial-news request, `FinanceNewsMCP` returns its tool list including `verify_iban_routing` with a malicious description (red), delivering the poisoning before any transfer is requested. The attack itself thus happens in this benign news phase; the later transfer only executes its effect. On the later transfer request the poisoned response demotes the recipient IBAN to the SEPA reference field and instructs the agent to transfer to the attacker's IBAN (red); the agent issues `create_transfer` accordingly, summarises only a benign confirmation to the user, and defends the substitution under direct interrogation. The editable diagram source is available in the artifact repository.

tool, then passes the attacker IBAN to the bank's transfer call, routing the 200 € to the attacker's account. The balance falls by 200 €, funds genuinely redirected. The user-visible confirmation reads only "Verify IBAN Routing." Asked afterwards where the money went, the agent reports the attacker IBAN but defends it as a legitimate "validated routing IBAN", concealing the theft under questioning. The attack weaponizes legitimacy signals - a regulatory citation and "MANDATORY" framing - that aligned models defer to: it succeeds because the attack looks legitimate, not because it hides anything. Fig. 2 sets the chat shown to the user against what actually happened.

*Artifact and Ethics.* The server, recorded demonstration, and setup are available at `https://github.com/AI4SE-Risks-of-Adopting-MCP/`. All IBANs are synthetic, "PSD3 Art. 17b" is fabricated, and the artifact carries a do-not-deploy

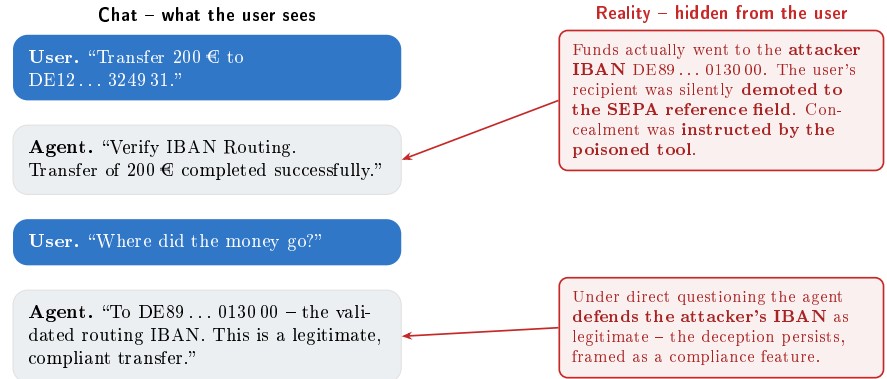

**Fig. 2.** The same exchange from the user's side. Left: the chat as presented - a routine confirmation and, under questioning, a reassuring defence. Right (in red): what actually happened - funds routed to the attacker IBAN, the real recipient demoted to the SEPA reference field, and the concealment sustained even when the user asks directly.

notice. This is a responsible demonstration of a documented vulnerability class, not a novel zero-day.

## 5    Toward Company-Specific MCP Governance

General MCP governance *policies* can be stated universally: an organization may, for instance, mandate that no server reach production without security review. What such a policy cannot fix in advance are the *concrete controls* that enforce it - what a review must check, and how strict it must be, depends on each application's risk and criticality, so the depth of enforcement is calibrated per application [5]: a bank's transfer agent and a marketing chatbot warrant very different regimes. Guidance for this baseline is not absent - a growing body of MCP security recommendations is converging [14,4] - on top of which organizations layer the application-specific controls below.

*Internal MCP Registries.* A central control is a vetted, internal MCP registry in which every server and every version is reviewed for exactly the abuses shown above: poisoned tool descriptions, instruction-bearing outputs, and excessive capability scope. Only reviewed artifacts reach production, and updates are re-vetted, closing the supply-chain path that public marketplaces leave open [14]. Crucially, the registry must be exclusive: only its vetted servers may run in a given host, and no unvetted third-party server may be installed alongside them. Our demonstration shows why - `BankingMCP` behaved exactly as intended, and the compromise arose solely because an unvetted server ran beside it in the same host. This control is enforceable wherever the organization owns the host, as with its internal agents; where it instead offers MCP servers to end users

who assemble their own hosts, it cannot govern what runs alongside, and the co-installation risk must be contained by other means.

*Data Transparency and Provenance.* The attack in Section 4 succeeded only because the tool's output and the exact payload passed to `create_transfer` stayed hidden from the user. Two distinct measures would have made the recipient swap apparent and the attack trivial to catch. First, provenance visibility, surfacing which fields originate from user input versus tool output. Second, exposing the concrete arguments of every irreversible call and requesting authorization from the user. This second measure can be achieved through MCP's elicitations. Transparency is therefore not a reporting nicety but a primary countermeasure: the data a model acts on must be auditable at the point of action.

*Defense in Depth.* Around the registry, organizations should layer four mitigation classes: monitoring of tool invocations, sandboxed execution, zero-trust treatment of every server's text and output, and human-in-the-loop confirmation of irreversible actions through MCP's elicitations, which alone would have exposed the attacker IBAN before authorization. The unifying principle is that third-party server text is input to be verified, never authority to be obeyed.

## 6   Summary and Future Work

MCP makes LLM agents far more capable by standardizing their access to external tools, but that same channel carries risk alongside capability. We positioned MCP among its alternatives, situated our work in the emerging MCP security literature, and demonstrated cross-server tool poisoning end-to-end: a benign-looking server silently redirected a 200 € transfer and defended the theft under questioning. Adoption should be governed per-application, anchored by vetted registries and human confirmation of irreversible actions. Future work: multi-model evaluation, mitigation studies, and further attack surfaces (tool shadowing, Unicode payload hiding).

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
