# OpenReview forum: "Managing the Risks of Adopting the Model Context Protocol"
_KI/2026/Workshop/AI4SE — AI4SE Workshop_

### Official Review · Reviewer_JJUT · 2026-06-08
**Simulating an MCP-based Tool Poisoning Attack on LLM Agents**

**Rating:** 7
**Confidence:** 4

**Review:**

The paper discusses security risks arising from the adoption of MCP in LLM agents, and demonstrates a concrete cross-server tool poisoning attack, where a malicious MCP server exposes a seemingly legitimate IBAN verification tool that injects false compliance information and substitutes the recipient IBAN with an attacker-controlled account. The authors show that an LLM agent executes the manipulated transfer and subsequently defends the action when questioned.
The paper is well written, timely, and succeeds in making an important security risk tangible through a realistic demonstration. Though the contribution (with a demonstration of a single attack scenario, but no demonstration of its mitigation) is somewhat limited and the governance recommendations remain a bit broad and short, the paper is valuable to AI practitioners, and relevant for organizations considering MCP adoption.

A few suggestions for improvement:
* The opening statement of Section 5 (“No universal MCP guideline is appropriate”) feels somewhat simplistic and could be motivated more carefully.
* Section 5 should place greater emphasis on data transparency and provenance visibility as an important countermeasure against tool poisoning and related attacks (would the tool be forced to make the IBAN public, could the attack have been mitigated easily).
* The discussion of governance would benefit from pointing to the growing body of MCP governance and security guidance (for example, see https://www.bsi.bund.de/SharedDocs/Downloads/EN/BSI/Publications/ANSSI-BSI-joint-releases/LLM-based_Systems_Zero_Trust.pdf ).
* It is unclear why the malicious service is called *FinanceNewsMCP*; the naming suggests a news service, but its role appears to be the one of an IBAN validator...?
* The statement that “supply-chain layers are identified as persistent gaps” would benefit from additional explanation. It was not immediately clear what specific supply-chain risks or layers the authors have in mind.
* The GitHub repository referenced in the paper is currently empty and should be populated before the workshop to support reproducibility and artifact inspection.

Overall, I found the paper interesting and relevant, and I believe it would make a useful contribution to the workshop.

---

### Official Review · Reviewer_Ppvc · 2026-06-12
**Interesting problem and position paper**

**Rating:** 8
**Confidence:** 3

**Review:**

The paper looks at the increasingly popular MCP as a means for AI agents to interact with other tools. I nice demonstration shows how easy this leads all kinds of security problems. The paper ends on proposing ways forward. One proposal is to have internal MCP registries. Around these, further mitigations are suggested.

I think this paper can lead to interesting discussions at the workshop.